# A Review of the Respiratory Health Burden Attributable to Short-Term Exposure to Pollen

**DOI:** 10.3390/ijerph19127541

**Published:** 2022-06-20

**Authors:** Nur Sabrina Idrose, Caroline J. Lodge, Bircan Erbas, Jo A. Douglass, Dinh S. Bui, Shyamali C. Dharmage

**Affiliations:** 1Allergy and Lung Health Unit, Melbourne School of Population and Global Health, The University of Melbourne, Melbourne, VIC 3053, Australia; nidrose@student.unimelb.edu.au (N.S.I.); clodge@unimelb.edu.au (C.J.L.); dinh.bui@unimelb.edu.au (D.S.B.); 2Centre for Food and Allergy Research, Murdoch Children’s Research Institute, Melbourne, VIC 3052, Australia; 3School of Psychology and Public Health, La Trobe University, Bundoora, VIC 3086, Australia; b.erbas@latrobe.edu.au; 4Department of Clinical Immunology and Allergy, Royal Melbourne Hospital, Parkville, VIC 3050, Australia; jdouglass@unimelb.edu.au; 5Department of Medicine, University of Melbourne, Melbourne, VIC 3052, Australia

**Keywords:** asthma, allergic rhinitis, pollen, COPD, COVID, respiratory health

## Abstract

Respiratory diseases such as asthma, allergic rhinitis (AR) and chronic obstructive pulmonary disease (COPD) affect millions worldwide and pose a significant global public health burden. Over the years, changes in land use and climate have increased pollen quantity, allergenicity and duration of the pollen season, thus increasing its impact on respiratory disease. Many studies have investigated the associations between short-term ambient pollen (i.e., within days or weeks of exposure) and respiratory outcomes. Here, we reviewed the current evidence on the association between short-term outdoor pollen exposure and thunderstorm asthma (TA), asthma and COPD hospital presentations, general practice (GP) consultations, self-reported respiratory symptoms, lung function changes and their potential effect modifiers. The literature suggests strong evidence of an association between ambient pollen concentrations and almost all respiratory outcomes mentioned above, especially in people with pre-existing respiratory diseases. However, the evidence on sub-clinical lung function changes, COPD, and effect modifiers other than asthma, hay fever and pollen sensitisation are still scarce and requires further exploration. Better understanding of the implications of pollen on respiratory health can aid healthcare professionals to implement appropriate management strategies.

## 1. Introduction

### 1.1. Outdoor Pollen Is a Risk Factor for Respiratory Diseases

Pollen, a type of aeroallergen, is an established risk factor for allergic rhinitis (AR) and asthma [1,2,3], but evidence for a relationship with chronic obstructive pulmonary disease (COPD) is still emerging. Pollen can trigger allergic inflammation which is IgE-dependent, but non-allergic inflammatory changes may still be possible [4]. Due to their micronic sizes, pollen can easily enter the respiratory tract during inhalation. Intact pollen grains are usually >20 μm in size and deposit in the upper respiratory tract [5], but pollen fragments can be less than 1 μm in size and therefore can deposit in the lower airways [1]. Subsequently, this can trigger a cascade of immune responses, leading to adverse respiratory effects. These adverse respiratory effects can result in asthma and COPD hospital presentations, general practice (GP) consultations, self-reported symptoms, or subtle lung function changes. Because pollen exposure can be over the life course, from pre-natal, post-natal, childhood through to adulthood, it is important to understand its impact, so that necessary measures such as behavioural modification and environmental planning can be adopted to reduce the burden of exposure.

### 1.2. What Do We Know about Pollen?

Pollen is the male gamete of seed-bearing plants, comprised of Angiosperms and Gymnosperms. It is a fine, dust-like substance that can be transported by several methods, including biotic (insect-borne) and abiotic (wind) vectors. The prominent allergenic pollen families are grass, tree and weed. These pollens are produced by wind-pollinated plants and are adapted to have excellent aerodynamic properties, allowing distribution over hundreds of kilometres [6,7]. The primary allergenic pollen differs between countries and regions, due to geographical variations [8,9,10]. For example, the main allergenic pollen family in the Southern Mediterranean is nettle weed [11], but grass is prominent in other temperate regions such as northern Europe and Australia [12]. Temperate grasses (e.g., *Pooideae*) dominate in cooler climates, while subtropical grasses (e.g., *Panicoideae* and *Chloridoideae*) dominate in subtropical climates [13]. Although allergenic pollen only account for approximately 20–30% of the total annual pollen load, they are abundant during the species’ flowering seasons [14], thereby triggering symptoms.

Climate change can affect pollen species’ distribution and phenology. Over the years, changes in land use and climate, such as rising temperatures and carbon dioxide levels, have led to increased pollen quantity, allergenicity and duration of the pollen season [15,16,17]. In some instances, during thunderstorm activity, pollen can burst into hundreds of sub-pollen particles of paucimicronic size, causing a phenomenon called “thunderstorm asthma” (TA). Scientists predict that climate change will increase the concentration of airborne pollen, as the frequency and intensity of rainfall is also likely to increase [18,19]. As a result, the prevalence and severity of pollen-induced asthma, AR and sensitisation are also expected to increase [19,20,21].

### 1.3. Proposed Immunological Mechanisms Triggered by Pollen Exposure

Experimental challenge studies have indicated that pollen exposure can induce type I hypersensitivity, that is, early- (within minutes) and late- (within 8–12 h of exposure) phase effects on the airways, predominantly type-2 inflammation [22,23]. In our recently published systematic review of observational studies, we found substantial evidence to support that outdoor pollen exposure was associated with type-2 inflammation such as eosinophils, IL-4 mediated through IgE binding to mast cells, in both the upper and lower airways, especially in those with asthma and/or seasonal AR [24]. Individuals with pre-existing allergic sensitisation may be more susceptible to allergens due to increased sensitivity as the season progresses (i.e., ‘priming’) [25,26] and/or the fact that they could have leaky mucous barriers due to underlying inflammation [27]. We also found some evidence of innate inflammatory response to pollen, i.e., neutrophilic and macrophage-mediated, but to a lesser extent. Very little evidence was found for an association with non-specific or type-1 inflammatory responses such as IFN-γ and IL-6.

Inflammatory response to pollen is suggested to peak within 72 h of exposure, as proposed by studies involving asthma hospital presentations [8]. Because only two studies in our review examined lagged effects of pollen on airway inflammation, we subsequently evaluated these associations using data collected from the Melbourne Atopy Cohort Study (MACS), which consisted of primarily ‘high-risk’ adults, defined by having at least one first-degree family member with a history of allergic disease [28]. We found increased grass pollen concentrations to be associated with increased eosinophilic airway inflammation (measured by the fractional exhaled nitric oxide, FeNO) 1–2 days after exposure (i.e., at lag 1 and lag 2) [29]. Lambert and colleagues [30] also assessed similar associations, but used data on adolescents from two German birth cohorts, the GINIplus and LISA. They observed the strongest associations between grass pollen concentrations and FeNO to be on the same day (lag 0) and lag 1.

These lagged responses of up to days indicate that pollen may also induce type IV hypersensitivity, which is often linked with a more delayed immune response of 24 to 72 h of exposure [31]. However, further investigation is required because this mechanism is still poorly understood due to temporal and geographical differences in pollen allergenicity, as well as physiology differences between individuals [32]. It is also possible that children and adults have different types of hypersensitivity reactions due to different behavioural lifestyle events.

## 2. Aims of This Review and Search Strategy

In this review, we summarise the current evidence on the association between short-term outdoor pollen (defined as the impact within days or weeks of exposure) and respiratory health, ranging from the highest to lowest burden on health services in terms of patient demand, i.e., thunderstorm asthma (TA) events, asthma and COPD hospital presentations, general practice (GP) consultations, self-reported respiratory symptoms and lung function changes (Figure 1). In doing so, we also list the effect modifiers that have been assessed in the included studies, to inform high-risk groups and periods (Figure 2). To the best of our knowledge, there has not been a review of studies investigating the impacts of short-term outdoor pollen exposure on respiratory health burden and their effect modifiers. Previously published reviews have focused on a single respiratory outcome, e.g., asthma hospitalizations in children [33], rather than a range of different respiratory health measures.

We used a systematic search strategy to identify articles on the PubMed electronic database. First, we gathered all available systematic reviews on pollen and respiratory health. If a systematic review was available on any of the outcomes of interest, we then used narrow search terms to identify recent original articles published in the last two years (i.e., between 1st of January 2020 to 21st January 2022) to supplement the review findings. If no systematic review had been published, all relevant original articles from inception until 21st of January 2022 were included, where relevant. These were applicable to the following outcomes, as they have not yet been systematically reviewed: adulthood asthma and COPD hospital presentations and GP consultations. The search terms used can be found in Appendix A. All relevant papers were narratively synthesised and grouped into relevant subheadings.

## 3. Short-Term Impact of Pollen on Adverse Respiratory Outcomes

### 3.1. Thunderstorm Asthma (TA)

Grass pollen plays a crucial role in thunderstorm asthma (TA), as evidenced by our systematic review [34]. TA is defined by acute bronchospasm that is triggered within minutes or hours following a thunderstorm or convergence line weather event (defined as a band of rain and cloud, formed when winds from opposite directions collide) with or without the presence of lightning [35,36]. TA can perpetuate an early asthmatic response that is severe enough to seek emergency treatment and, in some instances, death [35,37]. It can cause an unforeseen and sudden surge in respiratory-related GP and hospital presentations, therefore imposing a considerable burden on health services. Airway inflammation during TA is characterised by mucus production, mucosal edema and IgE-mediated mast cell degranulation, followed by increased sputum eosinophil cationic protein (ECP), sputum eosinophils and IL-5-positive cells [38]. These changes caused by TA are similar to asthma, which is why it has the same ICD code (J45, J45.0, J45.1, J45.8, J45.9 and J46) [39]. TA effects were less severe in those who had no previous diagnosis of asthma or hay fever, such that the majority did not require emergency medical assistance [40,41]. Nonetheless, this reveals a hidden at-risk population and an urgent need for early warning systems where, delivery of real-time pollen information is readily accessible by at-risk communities.

Seasonal peaks in asthma admissions have been reported during spring so TA might only be the worst of a seasonal phenomenon [42]. Of the 20 studies included in our systematic review (up until 15 April 2019), 15 presented some evidence of an association between pollen and TA, with nine demonstrating lagged effects of up to four days [34]. Elliot and colleagues [35], who investigated the recent 2021 TA event in England (and therefore was not included in our review), also reported high grass pollen levels at the time of the occurrence. Because these events usually occurred during pollen seasons, it is hypothesised that there was a priming phenomenon or a prior sensitisation phase, in which people were already exposed and usually pollen-sensitised [24,43].

During a thunderstorm, whole pollen grains (≥20 μm) are disseminated up and horizontally into the air. The convergence line weather events, plus a sudden drop in temperature and air pressure, high humidity, electrical ions, lightning strikes and heavy rainfall, facilitate the rupturing of whole pollen grains into Lol p 5-enriched sub-pollen particles (≤2.5 μm) [36,44,45,46]. The gusty wind then transports the sub-pollen particles over long distances, while the cold downdraft and outflows re-deposit the ruptured pollen grains onto or near to the ground [46,47]. As the sub-pollen particles are several times smaller than intact pollen grains, they can evade filtration by the nasopharynx and penetrate deeper into the airways, provoking primed individuals, even those with no history of diagnosed asthma, to have a more dramatic, asthmatic response [48]. Nonetheless, pollen is not the only factor for TA onset. Other aeroallergens such as fungi can also play a part [49].

Thunderstorm asthma prediction models should consider the role of air quality and its potential synergistic effect with aeroallergens. Darvall et al. [50] demonstrated a sharp increase in PM_10_ concentrations coinciding with the storm front, and although this suggests that more PM_10_ could be stirred up in the winds during a thunderstorm, it could also indicate that ruptured aeroallergens are mistaken for particulate matter (PM) and therefore, are undetectable by volumetric spore traps [34]. To support this, most observational case report studies in our systematic review detailed high intact pollen concentrations in the days preceding the thunderstorm, but the concentrations were low–moderate during the event [34].

### 3.2. Asthma and COPD Hospital Presentations

#### 3.2.1. Childhood Asthma

To date, three systematic reviews [8,13,33] have evaluated the relationship between ambient pollen and childhood asthma hospital attendances, in which ambient grass and birch pollen were reported to be important risk factors, but only in non-subtropical climates.

Erbas et al. [8], who examined asthma emergency department (ED) presentations, included 14 eligible studies and reported an increased risk of 1% to 14% of ED presentations associated with increasing ambient pollen exposure. There was a minimum threshold of 10 grains/m^3^ and some studies demonstrated that the effect flattened after reaching a certain threshold, ranging from 20–50 grains/m^3^. This threshold could be population- or location-specific, as it reflects the variation in pollen sensitisation rates, pollen allergenicity and fewer data points at higher pollen concentrations [32,51,52]. Meta-analysis was only possible for grass pollen, demonstrating a 1.88% (95% CI = 0.94%, 2.82%, I^2^ = 0%; *n* = 3) increase in number of asthma ED presentations for every 10 grains/m^3^ increase at lag 3.

Shrestha et al. [33], who examined asthma hospitalisations, included 12 eligible studies and demonstrated a relationship with ambient grass and birch pollen. For every 10 grains/m^3^ increase in grass pollen at lag 0 and birch pollen at lag 2 and lag 0–6, there was a 3% (95% CI = 1%, 4%, I^2^ = 0%; *n* = 2) and 0.85% (95% CI = 0.4%, 1.3%, I^2^ = 0%; *n* = 2) increase in asthma admissions, respectively.

Simunovic et al. [13], who assessed asthma ED presentations and hospitalisations in subtropical climates, reported little to no evidence of an association with grass pollen on the same day or lagged in the six studies that assessed children only. A meta-analysis was not possible for this review. In subtropical regions, grass pollen seasons are longer in duration with multiple peaks and there are probably many different grass pollen species involved, but in temperate regions, the seasons are shorter with a single, relatively higher peak and only one or two dominant grass pollen species [12,53]. This could explain why a strong association could not be detected in the subtropics.

More recent studies from various regions have reported increasing grass [54], tree [54,55,56] and weed [54,55] pollen concentrations to be associated with an increased risk of asthma exacerbations. Of these three studies, two were conducted in temperate climates [54,55] and one was in a subtropical climate [56].

Only one study investigated the role of ambient pollen concentrations on asthma readmissions. Vicendese et al. [57] observed higher rates of readmission within 28 days in boys during the temperate grass pollen season. The authors explained that this could be because boys were more sensitive to pollen, and this may have led to severe asthma reactions that required multiple admissions. Alternatively, it may suggest poorer adherence to treatments in boys, as they accounted for 60% of all admissions.

#### 3.2.2. Adulthood Asthma

Similarly, ambient pollen is an important risk factor for adulthood asthma, but only in non-subtropical climates. Only one systematic review examined the role of outdoor short-term pollen exposure on adult asthma hospital attendances, but it focussed on subtropical climates. Comparable to what was found in children, Simunovic et al. [13] reported little to no evidence of an association between grass pollen (on the same day or lagged) and hospitalisations in the two studies that assessed adults only.

Studies performed in non-subtropical climates have not yet been systematically reviewed. We identified 14 original studies investigating such relationships from the search. Of these, 13 reported some evidence of an association between increasing ambient pollen concentrations and adult asthma hospital attendances [20,55,58,59,60,61,62,63,64,65,66,67,68]. For example, Osborne et al. [66], who conducted a study in London, UK, observed increasing grass pollen concentrations at lag 4 and lag 5 to be associated with increased asthma admission rates. When classified into pollen ‘alert’ levels based on the UK’s Met Office, the authors found ‘very high’ pollen days (vs. low) to be associated with increased admissions 2 to 5 days after, peaking at lag 3 (Incidence rate ratio = 1.45 [95% CI = 1.2, 1.78]) [66]. Although this study did not find an association with tree pollen, others have [20,55,60,62,65]. Contrarily, weed pollen has been shown to be important in Hungary [64,68]. One study indicated a “threshold level”, as the 2nd quartile of pollen distribution imposed the maximum effect on asthma hospital admissions [59]. However, the authors stated that this could be because admissions occurred primarily in pollen-sensitised individuals, so once they were affected, the impact on those not sensitised may be lower. The study [69] that reported no evidence of a relationship failed to do extensive analysis with pollen, and only had 232 attendances recorded that year, so there could be insufficient power to detect associations. Moreover, Oulu is an industrial town, so pollen may not be as important there.

#### 3.2.3. Adulthood COPD

Very few studies investigated the relationship between outdoor pollen concentrations and COPD and this evidence has also not been systematically reviewed. Of the five studies that we discovered, two reported some evidence of an association [70,71], while others did not [62,72,73]. Brunekreef et al. [70] observed a small dose–response relationship between average weekly grass pollen concentrations and daily COPD mortality in the Netherlands. Compared to the lowest exposure category (<22 grains/m^3^), the moderate (22–77 grains/m^3^), high (78–135 grains/m^3^) and extreme (>135 grains/m^3^) exposure categories were associated with a 1.095 (95% CI = 1.05, 1.14), 1.12 (95% CI = 1.07, 1.18) and 1.15 (95% CI = 1.08, 1.23) increased risk of daily COPD mortality, respectively. These effect sizes were comparable to the association between air pollution and mortality [70]. Additionally, Hanigan et al. [71] showed that for every IQR increase in total pollen concentrations, daily COPD admissions in Darwin, Australia, increased by 33.2% (95% CI = 12.8, 57.3). However, the authors stated that replication of the study findings is needed as the study sample was relatively small (*n* = 334 over 20 months). Studies that reported no evidence of an association [62,72,73] did not provide effect estimates, so we could not determine the strength and confidence of the relationships.

### 3.3. General Practice (GP) Consultations

The relationship between ambient pollen concentrations and GP consultations also has not been systematically synthesised. Of the five studies that we found, all reported a positive association with GP consultations for AR [74,75,76,77] and asthma [75,76,78]. One study in Beijing, China, observed the association with AR consultations to be strongest at lag 0 [74] (RR = 2.6; 95% CI = 2.6, 2.7 for every 10 grains/m^3^ increase in total pollen). Interestingly, a UK study demonstrated that peak AR consultations coincided with peak grass pollen concentrations, but peak asthma consults occurred only 2–3 weeks later [75], indicating potential differences in disease mechanisms. Comparatively, Huynh et al. [78] observed a strong linear relationship between average weekly grass pollen concentrations and weekly asthma consults (RR = 1.54, 95% CI = 1.33, 1.79 for every IQR increase [i.e., 17.6 grains/m^3^]).

### 3.4. Self-Reported Respiratory Symptoms

In those with pre-existing allergic conditions, even low–moderate levels of pollen are likely to trigger respiratory symptoms on the same day of exposure. Kitinoja et al. [79] recently published a systematic review of 26 studies that assessed the association between pollen concentrations and self-reported respiratory symptoms in allergic and/or asthmatic subjects. Some studies were eligible for a meta-analysis, in which the authors reported a 7% (95% CI = 4%, 9%; *n* = 3; I^2^ = 28.7%) and 1% (95% CI = 0%, 2%; *n* = 6; I^2^ = 68%) increase in the risk of self-reported upper and lower respiratory symptoms, respectively, for every 10 grains/m^3^ increase in any pollen. However, all studies included in this systematic review investigated pollen exposure at lag 0 only and there was moderate–high heterogeneity.

The most recent study evaluated the relationship between daily ambient pollen concentrations and respiratory symptoms logged by users of a smartphone application, called AirRater, in Tasmania, Australia [80]. There was a non-linear association of up to 3 days following exposure and no minimum threshold of pollen concentration, indicative of no ‘safe level’, akin to air pollution effects. The lag 0 association was the strongest with an RR of 1.31 (95% CI = 1.26, 1.37 at 50 grains/m^3^). Furthermore, more users reported symptoms in the upper respiratory tract compared to the lower respiratory tract, which is consistent with the systematic review by Kitinoja et al. [79]. It is important to note that most of the app users self-identified as having a history of asthma and/or AR, and they were also more likely to be health-literate compared to the public.

### 3.5. Lung Function Changes

There is little evidence that pollen season or pollen concentrations on the same day were associated with lung function changes, but the four studies that investigated lagged responses reported some evidence of an association. These four studies also suggest that ambient pollen may be associated with different lung function measures, depending on age. It seems like allergenic pollen is more associated with the forced expiratory volume in 1 s (FEV_1_) and forced vital capacity (FVC) in children, and the FEV_1_/FVC ratio and mid-forced expiratory flow (FEF_25–75%_) in adults.

Two systematic reviews have been published on this topic [24,79]. In our qualitative review of seven population-based studies that investigated such relationships [24], only the study that measured associations for lagged effects reported evidence of an association, with significant reductions in FEV_1_ and FVC in 8-year-old children with increasing grass pollen concentrations at lag 1, lag 0–3 and lag 0–7 [81]. In the quantitative review by Kitinoja and colleagues [79], they found no evidence of a relationship between ambient pollen on the same day and peak expiratory flow (PEF) (*n* = 2) and FEV_1_ (*n* = 2).

Subsequently, we performed a data analysis of the MACS high-risk cohort which consisted of primarily adults and observed increased grass pollen concentrations to be associated with middle–small airway changes 2–3 days after exposure, as reflected by the FEV_1_/FVC ratio and FEF_25–75%_, respectively [29]. Similarly, Lambert et al. [82] observed a reduction in FEV_1_ and FVC with increasing concentrations of tree pollen at lag 1 and lag 3 in 8-year-old ‘high-risk’ children residing in Sydney, Australia [82]. The same authors investigated similar relationships in adolescents of the GINIplus and LISA cohorts in Germany, but observed the association to be present only in those who were pollen sensitised [30]. Using an unsupervised approach, another recent study of the PARIS cohort demonstrated that children in the ‘grass pollen’ cluster (i.e., moderate grass pollen exposure and low air pollution exposure) had reductions in FEV_1_ and FVC, when compared to children in the ‘low exposure’ cluster (i.e., no pollen exposure and low air pollution exposure) [83].

## 4. Evidence on Potential Effect Modifiers of the above Associations

The adverse effects of short-term ambient pollen exposure on the respiratory tract have been shown to be more severe in certain groups of people and/or environmental conditions. The effect modifiers that have been investigated thus far are air pollution, residential greenness, asthma, AR, pollen sensitisation, adherence to medications, human rhinovirus (HRV) infection, food allergy, eczema, socioeconomic status (SES), age, sex and ethnicity (Figure 2).

### 4.1. Air Pollution

It is believed that air pollutants can enhance pollen allergenicity by altering its content and morphology [84], its production [85], and its immunomodulatory and chemotactic properties [86]. There is a growing body of epidemiological evidence suggesting a synergistic relationship of pollen and air pollutants on respiratory health, but the findings have been inconsistent.

To date, two systematic reviews have been published on this. Anenberg et al. [87] and colleagues systematically reviewed the interactive relationships of multiple environmental factors, such as temperature, air pollutants and pollen on respiratory morbidity and mortality. These respiratory morbidities included self-reported symptoms and hospital presentations. Of the 56 studies that were deemed eligible, six examined the interactive effects between pollen, air pollution, and heat, and 10 examined the synergistic associations of pollen and air pollution only. The authors concluded that there was inconsistent evidence of interaction, and the studies were of low quality, so no proper conclusion could be drawn. The other review by Lam et al. [88] was slightly different, but the conclusion was the same. The authors systematically reviewed 35 studies that examined the interactive relationships between outdoor allergens such as pollen and fungal spores and air pollutants, and respiratory outcomes. These respiratory outcomes included asthma, wheeze, lung function changes, GP consultations and self-reported respiratory symptoms. Similarly, the authors could not determine which allergens and air pollutants were more likely to interact with one another because the findings were too inconsistent. Meta-analysis was also not possible due to the high heterogeneity and there were limited studies with reported effect estimates.

The most recent study analysed a total of 36,996 symptom reports from 2272 users of a smartphone app and documented an almost doubled risk of self-reported respiratory symptoms associated with ambient pollen concentration on high PM_2.5_ days (at 50 grains/m^3^, RR = 1.54, 95% CI = 1.42,1.66 for PM_2.5_ level centred at 100 μg/m^3^ vs. RR = 1.26, 95% CI = 1.2, 1.33 for PM_2.5_ level at baseline) [80]. However, the *p*-values for the interaction term were not reported here.

### 4.2. Residential Greenness

Lambert et al. [30,82] performed two studies investigating effect modification by residential greenness for the association between ambient pollen concentration and airway inflammation. Both studies found evidence of interaction, such that those living in areas of higher residential greenness were at-risk of increased airway inflammation with increasing ambient pollen concentrations. However, only one of the studies reported clinically meaningful effect modification (e.g., FeNO = 0.12%, 95% CI = 0.05, 0.18 for every 15 grains/m^3^ increase in birch pollen at lag 1 in high residential greenness areas), possibly because this study had more statistical power [30].

### 4.3. Asthma, Allergic Rhinitis (AR) and Pollen Sensitisation

The impact of short-term outdoor pollen exposure on respiratory health in those with asthma [83], AR [56] or pollen sensitisation [30,89] has been extensively studied and established. Recent studies, including our own, have supported this by showing stronger associations between ambient pollen levels and lung function in those with asthma [29,83], AR [29] and/or pollen sensitisation [29,30].

### 4.4. Adherence to Preventer Medications

TA epidemics have shown that those with doctor-diagnosed asthma, plus poor adherence to preventer medications, were especially at risk of ICU admissions and mortality [46,50]. To support this, one study found that children with asthma who had poor adherence to anti-inflammatory medications had increased odds of asthmatic episodes with increasing pollen concentrations [90].

### 4.5. Human Rhinovirus Infection (HRV)

Erbas et al. [53] investigated effect modification by HRV for the association between outdoor grass pollen concentrations and incidence of childhood asthma admissions. The authors demonstrated that the risk of asthma admissions increased in boys with current HRV infection (OR = 1.42; 95% CI = 1.11,1.64 for every 50 grains/m^3^ increase). These findings were somewhat supported by those of Murray et al. [91], where they found children exposed to both allergens and HRV had an increased risk of asthma admission, although allergen exposure was not specific to grass. A possible mechanism is the “two-hit hypothesis”, whereby viral infections and allergens can increase the risk of asthma attacks by priming the respiratory mucosa and making it leakier. Nevertheless, further exploration is needed to understand these biological mechanisms.

### 4.6. Food Allergy

One study noted that food-sensitised girls had an increased risk of admission associated with increasing ambient pollen exposure (OR = 1.52; 95% CI = 1.03,2.24 for every 50 grains/m^3^ increase) [53]. Recently, we evaluated potential effect modification by food sensitisation and allergy status in 6-year-old children of the HealthNuts cohort [92]. Our analysis showed significant reversible obstructive lung function deficits with increasing grass pollen concentrations at all lags in children with food allergies (e.g., at lag-2, FEV_1_/FVC z-score = −0.50 [95% CI: −0.80, −0.20]; FEF_25–75%_ z-score = −0.40 [−0.60, −0.04]; bronchodilator responsiveness (BDR) = (31 [−0.005, 62] ml, per 20 grains/m^3^ pollen increase), but not in those who were food-sensitised tolerant or non-sensitised nor allergic. However, we could not tease out if these children also had asthma, as there was insufficient power to examine multiple interactions among allergic diseases that often coincide within an individual.

### 4.7. Eczema

We also examined potential effect modification by eczema in the HealthNuts study [92]. Children with current eczema (symptoms in the past 12 months or positive SCORAD score) had modest increases in BDR (18 [−8.0, 44] mL) for every 20 grains/m^3^ increase in grass pollen concentrations at lag 2, but not in children without. However, another study found no evidence of interaction between tree pollen and eczema diagnosis when investigating childhood asthma hospital presentations [56]. Collectively, these findings indicate that the impact of short-term pollen exposure on lung health in children with eczema may be subtle and not severe enough to cause asthma hospital presentations.

### 4.8. Socio-Economic Status (SES)

People with low SES may be more susceptible to pollen-induced asthma exacerbations because of reduced access to healthcare, lower air quality and lack of health literacy. Cakmak et al. [93] investigated the influence of SES on ambient pollen concentrations and asthma hospitalisations in Canada. The authors reported that the association was only present in the lowest education group, but not the highest, indicating that those with low SES may be more vulnerable to ambient pollen exposure. Conversely, De Roos et al. [56] observed no evidence of effect modification by SES in Pennsylvania, USA. This discrepancy in findings could be due to the differences in SES measure. The Canadian study measured SES using education levels, whilst the US study used payment source (public payer vs. other) as a proxy measure for SES. Education level is more widely used in research, thus making it a more accurate and established measure for SES [94].

### 4.9. Age

The literature suggests that people of all ages (except those aged 60 years and over) are susceptible to ambient pollen exposure. For example, Elliot et al. [35] observed notable spikes in asthma hospital presentations in all age groups in the recent 2021 TA epidemic. However, studies in the USA reported that ambient tree pollen induced more asthma hospital presentations and over-the-counter allergy medication sales in children compared to adults, potentially because children have smaller airways and a bigger risk of respiratory decompensation [20,95]. De Ross and authors who investigated asthma exacerbations in children (defined by asthma diagnosis code plus prescription of systemic steroid during the same visit) in relation to tree pollen exposure found that the risk increased across age groups, with the lowest being in those aged less than 2 years and highest being in those aged 12–18 years [56].

To support that susceptibility may increase with age, our meta-analysis showed that adults were more susceptible to airway inflammation with higher levels of eosinophil cationic proteins (ECP) and numbers of eosinophils, compared to children and adolescents, in the pollen season vs. out of the pollen season [24]. This is consistent with the age distribution of AR [46], implying that the allergen-specific IgE level is at its peak during adulthood [37]. We have also assessed interaction with age using the MACS cohort and found that adults aged at least 18 years had lower lung function and increased airway inflammation when exposed to increasing levels of pollen, but children and adolescents did not. Other studies have also shown that grass pollen only had an impact on asthma admissions in those aged 15 years and over [60,61]. Besides that, TA events have demonstrated that the mean age of those affected was 32 years [46] and that there was little to no change in childhood asthma presentations [48,96,97] (compared to daily average during similar periods).

Nonetheless, those aged 60 years and above were consistently reported to be less affected by ambient pollen compared to the other age groups [20,55,60,64,74,75], potentially because of (1) reduced pollen exposure as a result of reduced mobility and activity, (2) differences in disease management, (3) underdiagnosis due to other co-morbidities and (4) reduced prevalence of atopy and allergic respiratory diseases at >60 years old [98].

### 4.10. Sex

The literature implies higher prevalence of allergic diseases in boys compared to girls before puberty, but the opposite post-puberty [99]. Most studies [33,54,57] have indicated higher risk of childhood asthma hospital presentations in boys associated with ambient pollen concentrations. Only one study [56] reported no evidence of effect modification by sex, but 60% of the study sample were boys. Consistent with the literature, some studies [68,74] demonstrated that female adults had a higher risk of respiratory ED visits and GP consults in relation to ambient pollen concentrations, compared to male adults.

### 4.11. Ethnicity

To the best of our knowledge, no study has investigated specific gene–pollen interactions with regards to respiratory health. However, this is a research area of emerging interest because there is a growing body of evidence to show that overseas-born Asian individuals have an increased susceptibility to atopy and TA, with up to 5.4-fold increased odds of hospital presentations, but locally born Asian individuals do not [46,100]. Furthermore, black, non-Hispanic children had a higher risk of asthma exacerbations upon tree pollen exposure, but white, non-Hispanic children did not [56]. Besides genetics, other mechanisms that could explain this immune intolerance are varying sociodemographic factors, altered microbiome associated with hygiene hypothesis and vitamin D deficiency due to sun exposure and skin pigmentation [46].

## 5. Conclusions and Knowledge Gaps

In summary, we found strong evidence that increasing short-term outdoor pollen exposure is detrimental to respiratory health, similar to air pollution effects, especially to those with pre-existing allergic conditions. However, there are several knowledge gaps to consider. Potential effect modifiers such as air pollution and genetics require further exploration and more studies are needed to determine the influence of pollen on sub-clinical lung function and inflammation changes and COPD. Some topics such as adult asthma and COPD hospital attendances and GP consultations have not yet been systematically reviewed. Additionally, large time-series studies are necessary to detect associations and synergistic effects over wider populations. It is believed that the effect of pollen is broader than previously thought because there is evidence that it can trigger both allergic (e.g., allergic rhinitis or asthma) and non-allergic (e.g., COPD) inflammation. Hence, research on its implications on respiratory outcomes and also other non-respiratory outcomes, such as mental health, food allergy, eczema, cardiovascular and inflammatory diseases (e.g., rheumatoid arthritis) continues to be of great interest and importance.

## Figures and Tables

**Figure 1 ijerph-19-07541-f001:**
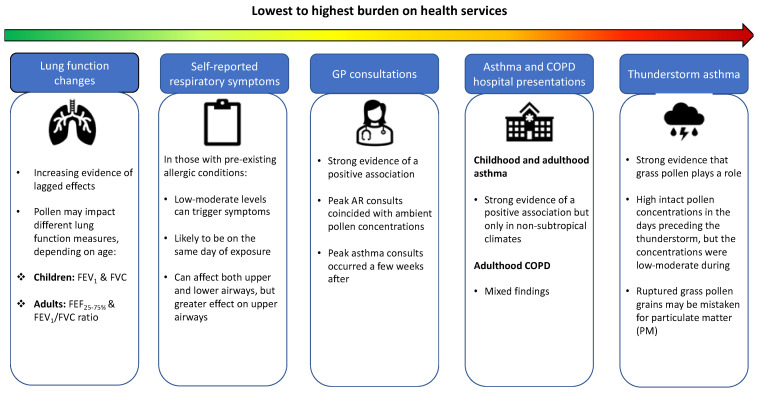
Summary of findings on the short-term impact of ambient pollen on respiratory outcomes (lowest to highest burden on health services).

**Figure 2 ijerph-19-07541-f002:**
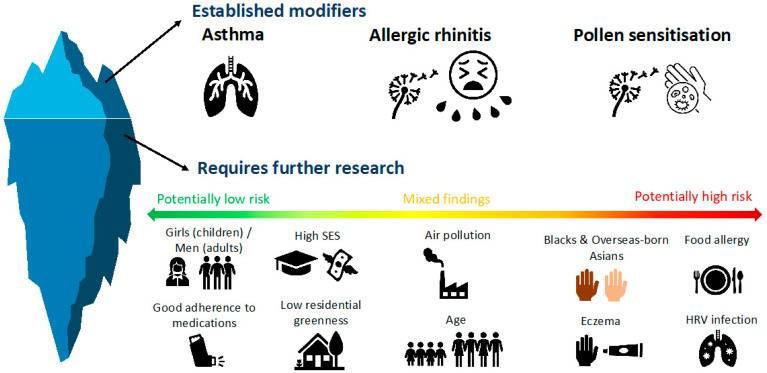
Current evidence of potential effect modifiers of the associations between short-term ambient pollen exposure and respiratory outcomes.

## Data Availability

Not applicable.

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
