# Peer review of "A Review of the Respiratory Health Burden Attributable to Short-Term Exposure to Pollen"

_ijerph, 2022, doi:10.3390/ijerph19127541_

Round 1

Reviewer 1 Report

This review paper summarized the current findings on pollen exposure and multiple respiratory health outcomes. This topic is important given both pollen and respiratory health outcomes are highly prevalent. However, this review article contained too much unnecessary information, which makes this article not easy to follow. And the discussion about COVID-19 seems overstated and not necessary. Here are my specific comments:

The first several paragraphs in the Introduction focus on introducing respiratory health outcomes. It would be better to directly jump into the association between pollen and respiratory health outcomes.

In the first sentence of “2. Aims of this review and search strategy”, how to determine which respiratory outcome is the “most severity” and which is the “least severity”?

Section 3.1. Is there an ICD code for thunderstorm asthma?

Another overall comment: please don’t use the results of review articles to support your statement because you are doing a review article. You need go to mention and cite the original research articles.

The paragraph starting with “With only 27 TA events” seems unrelated to pollen. The discussion about TA and COVID-19 makes the article disorganized.

Author Response

We thank the reviewer for their time in reviewing this paper. Please see the attachment for our response to comments. 

Reviewer 2 Report

Reviews the association between short-term outdoor pollen exposure and respiratory health, including thunderstorm asthma, asthma and COPD hospital presentations, GP consultations, self-reported respiratory symptoms, lung function changes, and their potential effect modifiers. This is a very large review of the current literature, including in-depth descriptions of mechanisms in the background section and discussion of potential effect modifiers. Overall the review is comprehensive and well-written. Overall the review seems rather lengthy. It appears to attempt to capture all evidence to date on the health impacts of pollen, including mechanism. While this may be helpful, the authors should consider if there is benefit with more brevity or focus of the paper. I only have minor comments:

Minor Comments

1.     The introduction is quite lengthy, perhaps inappropriately so. It’s almost a review within itself on the biologic mechanism of the impact of pollen on respiratory health. The authors may consider rethinking the background section or having this be its own review. It’s more difficult for the reader to understand why this review is needed (i.e., what’s the gap this manuscript is filling?).

2.     COVID-related component may be too much. This almost seems like a paper within itself (see description p. 4). Some of these items seem outside the scope of the paper.

3.     Figure 1 arrow is bidirectional, but the text indicates least to most clinically severe. Some clarification would be helpful. Also, is TA more clinically severe than COPD?

4.     The reader may be unfamiliar with TA and what the definition of a TA “event” is. More clarity in this would be helpful. Additionally, are there other reasons besides pollen that TA happens?

5.     Overall “short-term” is not defined clearly in the manuscript. This would be helpful.

Author Response

(The authors gave the same response as above.)

Round 2

Reviewer 1 Report

I don't have additional comments. Good work.